# SPREAD DIVERGENCE

## ABSTRACT

For distributions $p$ and $q$ with different supports, the divergence $\mathrm{D}(p||q)$ may not exist. We define a spread divergence $\tilde{\mathrm{D}}(p||q)$ on modified $p$ and $q$ and describe sufficient conditions for the existence of such a divergence. We demonstrate how to maximize the discriminatory power of a given divergence by parameterizing and learning the spread. We also give examples of using a spread divergence to train and improve implicit generative models, including linear models (Independent Components Analysis) and non-linear models (Deep Generative Networks).

## 1 INTRODUCTION

A divergence $\mathrm{D}(p||q)$ (see, for example Dragomir (2005)) is a measure of the difference between two distributions $p$ and $q$ with the property

$$\mathrm{D}(p||q) \geq 0 \quad \text{and} \quad \mathrm{D}(p||q) = 0 \quad \Leftrightarrow \quad p = q \tag{1}$$

We are interested in situations in which the supports of the two distributions are different, $\operatorname{supp}(p) \neq \operatorname{supp}(q)$. An important class is the $f$-divergence, defined as

$$\mathrm{D}_f(p||q) = \mathbb{E}_{q(x)}\left[ f\left( \frac{p(x)}{q(x)} \right) \right] \tag{2}$$

where $f(x)$ is a convex function with $f(1) = 0$. A special case of an $f$-divergence is the well-known Kullback-Leibler divergence $\mathrm{KL}(p||q) = \mathbb{E}_{p(x)}\left[ \log \frac{p(x)}{q(x)} \right]$. By setting $p(x)$ to the empirical data distribution, maximum likelihood training of a model $q(x)$ corresponds to minimising $\mathrm{KL}(p||q)$. However, this divergence may not be defined since the ratio $p(x)/q(x)$ can cause a division by zero.

This is a challenge since popular implicit generative models (Mohamed & Lakshminarayanan (2016)) of the form $q(x) = \int \delta\left(x - g_\theta(z)\right) p(z) dz$ only have limited support. In this case, maximum likelihood to learn the model parameters $\theta$ is not available and alternative approaches to measure the difference between distributions such as Maximum Mean Discrepancy (Gretton et al. (2012)) or Wasserstein distance (Peyré et al. (2019)) are required.

## 2 SPREAD DIVERGENCE

From $q(x)$ and $p(x)$ we define new distributions $\tilde{q}(y)$ and $\tilde{p}(y)$ that have the same support[1]. Using the notation $\int_x$ to denote integration $\int (\cdot)\, dx$ for continuous $x$, and $\sum_{x \in \mathcal{X}}$ for discrete $x$ with domain $\mathcal{X}$, we define a random variable $y$ with the same domain as $x$ and distributions

$$\tilde{p}(y) = \int_x p(y|x)p(x), \qquad \tilde{q}(y) = \int_x p(y|x)q(x) \tag{3}$$

where $p(y|x)$ is 'noise' designed to 'spread' the mass of $p$ and $q$ such that $\tilde{p}(y)$ and $\tilde{q}(y)$ have the same support. For example, if we use a Gaussian $p(y|x) = \mathcal{N}\left(y|x, \sigma^2\right)$, then $\tilde{p}$ and $\tilde{q}$ both have support $\mathbb{R}$. We also impose an additional requirement on the noise $p(y|x)$, namely that $\mathrm{D}(\tilde{p}||\tilde{q}) = 0 \Leftrightarrow p = q$. As we show in section(2.1) this is guaranteed for certain 'noise' distributions. Given these requirements, we can define the Spread Divergence $\tilde{\mathrm{D}}(p||q) \equiv \mathrm{D}(\tilde{p}||\tilde{q})$. This satisfies the divergence requirement $\tilde{\mathrm{D}}(p||q) \geq 0$ and $\tilde{\mathrm{D}}(p||q) = 0 \Leftrightarrow p = q$.

---

[1]For simplicity, we use univariate $x$, with the extension to the multivariate setting being straightforward.

For example, given two delta distributions $p_0(x) = \delta(x - \mu_0)$, $p_1(x) = \delta(x - \mu_1)$, the KL divergence (or $f$-divergence) between them is not defined. However, the spread KL divergence (or $f$-divergence) is defined. Assume a Gaussian noise distribution $p(y|x) = \mathcal{N}\left(y|x, \sigma^2\right)$ where $\sigma^2 = 0.5$, the "spreaded" delta distributions have the form: $\tilde{p}_0(y) = \int_x \delta(x - \mu_0)\mathcal{N}\left(y|x, \sigma^2\right) = \mathcal{N}\left(y|\mu_0, \sigma^2\right)$, $\tilde{p}_1(y) = \int_x \delta(x - \mu_1)\mathcal{N}\left(y|x, \sigma^2\right) = \mathcal{N}\left(y|\mu_1, \sigma^2\right)$. Therefore, the spread KL divergence (with Gaussian noise) between two delta distributions is equivalent to the KL divergence between two Gaussian distributions with the same variance, which has closed form (see appendix(D) for a derivation):

$$\widetilde{\text{KL}}(p_0(x)||p_1(x)) = \text{KL}(\tilde{p}_0(y)||\tilde{p}_1(y)) = ||\mu_0 - \mu_1||_2^2. \tag{4}$$

It's worth noticing that in the case of two delta distributions, the spread KL divergence is equal to the squared 2-Wasserstein distance (see Peyré et al. (2019); Gelbrich (1990)).

## 2.1 NOISE REQUIREMENTS FOR A SPREAD DIVERGENCE

Our main interest is in using noise to define a new divergence in situations in which the original divergence $\text{D}(p||q)$ is itself not defined. For discrete variables $x \in \{1, \ldots, n\}$, $y \in \{1, \ldots, n\}$, the noise $P_{ij} = p(y = i|x = j)$ must be a distribution $\sum_i P_{ij} = 1$, $P_{ij} \geq 0$ and

$$\sum_j P_{ij}p_j = \sum_j P_{ij}q_j \quad \forall i \quad \Rightarrow \quad p_j = q_j \quad \forall j \tag{5}$$

which is equivalent to the requirement that the matrix $P$ is invertible. There is an additional requirement that the spread divergence exists. In the case of $f$-divergences, the spread divergence exists provided that $\tilde{p}$ and $\tilde{q}$ have the same support. This is guaranteed if

$$\sum_j P_{ij}p_j > 0, \quad \sum_j P_{ij}q_j > 0 \quad \forall i \tag{6}$$

which is satisfied if $P_{ij} > 0$. In general, therefore, there is a space of noise distributions $p(y|x)$ that define a valid spread divergence. For example, the 'antifreeze' method of Furmston & Barber (2009) is a special form of spread noise to define a valid KL divergence (see also Barber (2012)).

For continuous variables, in order for $\tilde{\text{D}}(p||q) = 0 \Rightarrow p = q$, the noise $p(y|x)$, with $\dim(Y) = \dim(X)$ must be a probability density and satisfy

$$\int p(y|x)p(x)dx = \int p(y|x)q(x)dx \quad \forall y \in Y \quad \Rightarrow p(x) = q(x) \quad \forall x \in X \tag{7}$$

In the following section we discuss the special case of stationary noise for continuous systems.

## 3 STATIONARY SPREAD DIVERGENCES

Consider stationary noise $p(y|x) = K(y - x)$ where $K(x)$ is a probability density function with $K(x) > 0$, $x \in \mathbb{R}$. In this case $\tilde{p}$ and $\tilde{q}$ are defined as a convolution

$$\tilde{p}(y) = \int K(y - x)p(x)dx = (K * p)(y), \quad \tilde{q}(y) = \int K(y - x)q(x)dx = (K * q)(y) \tag{8}$$

Since $K > 0$, $\tilde{p}$ and $\tilde{q}$ are guaranteed to have the same support $\mathbb{R}$. A sufficient condition for the existence of the Fourier Transform $\mathcal{F}\{f\}$ of a function $f(x)$ for real $x$ is that $f$ is absolutely integrable. Since all distributions $p(x)$ are absolutely integrable, both $\mathcal{F}\{p\}$ and $\mathcal{F}\{q\}$ are guaranteed to exist. Assuming $\mathcal{F}\{K\}$ exists, we can then use the convolution theorem to write

$$\mathcal{F}\{\tilde{p}\} = \mathcal{F}\{K\}\mathcal{F}\{p\}, \quad \mathcal{F}\{\tilde{q}\} = \mathcal{F}\{K\}\mathcal{F}\{q\} \tag{9}$$

Let $\mathcal{F}\{K\} \neq 0$ or $\mathcal{F}\{K\} = 0$ on at most a countable set. Then

$$\mathcal{F}\{K\}\mathcal{F}\{p\} = \mathcal{F}\{K\}\mathcal{F}\{q\} \Rightarrow \mathcal{F}\{p\} = \mathcal{F}\{q\}. \tag{10}$$

The proof is given in appendix(A). Using this we can write

$$\text{D}(\tilde{p}||\tilde{q}) = 0 \Leftrightarrow \tilde{p} = \tilde{q} \tag{11}$$

$$\Leftrightarrow \mathcal{F}\{K\}\mathcal{F}\{p\} = \mathcal{F}\{K\}\mathcal{F}\{q\} \tag{12}$$

$$\Leftrightarrow \mathcal{F}\{p\} = \mathcal{F}\{q\} \Leftrightarrow p = q, \tag{13}$$

where we used the invertibility of the Fourier transform. Hence, for stationary noise $p(y|x) = K(y - x)$, we can define a valid spread divergence provided (i) $K(x)$ is a probability density function and (ii) $\mathcal{F}\{K\} \neq 0$ or $\mathcal{F}\{K\} = 0$ on at most a countable set. Interestingly, the sufficient conditions for defining a valid spread divergence such that $D(\tilde{p}||\tilde{q}) = 0 \Leftrightarrow p = q$ are analogous to the characteristic condition on kernels such that the Maximum Mean Discrepancy $\text{MMD}(p, q) = 0 \Leftrightarrow p = q$, see Sriperumbudur et al. (2011; 2012); Gretton et al. (2012). As an example of such a noise process, consider Gaussian noise,

$$K(x) = \frac{1}{\sqrt{2\pi\sigma^2}} e^{-\frac{1}{2\sigma^2}x^2}, \quad \mathcal{F}\{K\}(\omega) = \frac{1}{\sqrt{2\pi\sigma^2}} \int_{-\infty}^{\infty} e^{i\omega x} e^{-\frac{1}{2\sigma^2}x^2} dx = e^{-\frac{\sigma^2\omega^2}{2}} > 0 \quad (14)$$

Similarly, for Laplace noise

$$K(x) = \frac{1}{2b} e^{-\frac{1}{b}|x|}, \quad \mathcal{F}\{K\}(\omega) = \sqrt{\frac{2}{\pi}} \frac{b^{-1}}{b^{-2} + \omega^2} > 0 \quad (15)$$

Since in both cases $K > 0$ and $\mathcal{F}\{K\} > 0$, Gaussian and Laplace noise can be used to define a valid spread divergence.

# 4 MAXIMISING DISCRIMINATORY POWER

From the data processing inequality (see appendix(B)), adding spread noise will always decrease the $f$-divergence $D_f(\tilde{p}(y)||\tilde{q}(y)) \leq D_f(p(x)||q(x))$. Intuitively, spreading out distributions makes them more similar. If we are to use a spread divergence to train a model using maximum likelihood (see section section(5)), there is the danger that adding too much noise may make the spreaded empirical distribution and spreaded model distribution so similar that it becomes difficult to numerically distinguish them, impeding training. It is useful therefore to define spread noise that maximally discerns the difference between the two distributions $\max_\psi D(\tilde{p}(y)||\tilde{q}(x))$ for spread noise $p_\psi(y|x)$ parameterised by $\psi$. In general we need to constrain the spread noise to ensure that the divergence remains bounded.

We discuss below two complementary approaches to adjust $p(y|x)$ during training. The first approach adjusts the dimension-wise correlations (this corresponds to adjusting the covariance structure for Gaussian $p(y|x)$) and the second forms a mean transformation. In principle, both approaches can be combined and easily generalized to other noise distributions, such as Laplace noise.

## 4.1 LEARNING COVARIANCE STRUCTURE

Learning the covariance adjusts the shape of noise centered around the original model manifold. When we maximize the divergence between two spreaded distributions $\max_\psi D(\tilde{p}(y)||\tilde{q}(x))$, the learned noise will discourage overlap between the two distributions. Hence, if the data $p$ and model $q$ lie on the same manifold, the noise will be orthogonal to the manifold.

In learning the Gaussian spread distribution $p(y|x) = \mathcal{N}(y|x, \Sigma)$, the number of parameters in the covariance matrix $\Sigma$ scales quadratically with the data dimension $D$. We thus define $\Sigma = \sigma^2 I + LL^\mathsf{T}$ where $\sigma^2 > 0$ is fixed (to ensure bounded spread divergence) and $L$ is a learnable $D \times R$ matrix with $R \ll D$. Calculating the log likelihood and sampling can then be performed efficiently using standard Woodberry identities, see appendix(J).

## 4.2 LEARNING THE MEAN TRANSFORM

Consider $p(y|x) = K(y - f(x))$ for injective[2] $f$ and stationary $K$. Then, we define

$$\tilde{p}(y) = \int K(y - f(x)) p_x(x) dx \quad (16)$$

Note that this is a valid spread divergence since, using change of variables,

$$\tilde{p}(y) = \int K(y - z) p_z(z) dz, \quad p_z(z) = p_x(f^{-1}(z))/J\left(x = f^{-1}(z)\right) \quad (17)$$

---

[2]Since the codomain of $f$ is determined by its range, injective indicates invertible in this case.

where $J$ is the absolute Jacobian of $f$. Hence, $D(\tilde{p}_y||\tilde{q}_y) = 0 \Leftrightarrow p_z = q_z \Leftrightarrow p_x = q_x$. Each injective $f_\phi$ gives a different noise $p(y|x)$, we can thus search for the best noise implicitly by learning $f_\phi$.

In our experiments we use the invertible residual network Behrmann et al. (2018) $f_\psi : \mathbb{R}^D \to \mathbb{R}^D$ with $f_\psi = (f_\psi^1 \circ \ldots \circ f_\psi^T)$ denotes a ResNet with blocks $f_\psi^t = I(\cdot) + g_{\psi_t}(\cdot)$. Then $f_\psi$ is invertible if the Lipschitz-constants $Lip(g_{\psi_t}) < 1$ for all $t \in \{1, \ldots, T\}$. Note that when using the spread divergence for training (see section(5.2.2)) we only need samples from $\tilde{p}(y)$ which can be obtained from equation 16 by first sampling $x$ from $p_x(x)$ and then $y$ from $p(y|x) = K(y - f(x))$; this does not require computing the Jacobian or inverse $f_\psi^{-1}$.

# 5 SPREAD MAXIMUM LIKELIHOOD ESTIMATION

Minimising the forward KL divergence between the empirical data distribution $\hat{p}(x)$ and a model $p_\theta(x)$ is equivalent to Maximum Likelihood Estimation (MLE) of the parameters $\theta$ of the model. Minimising instead the forward spread KL divergence, $\widetilde{KL}(\hat{p}(x)||p_\theta(x)) = -\sum_{n=1}^N \log p_\theta(y_n) + const.$, where $y_n$ are sampled $i.i.d$ from $\tilde{p}(y) = \int_x p(y|x)\hat{p}(x)$, results in a new type of estimation, namely spread MLE. In what follows, we will discuss the statistical properties of spread MLE and demonstrate how it enables the training of models where maximum likelihood is not suited.

## 5.1 STATISTICAL PROPERTIES

Maximum likelihood is a cherished criterion because it exhibits many favourable statistical properties, mainly consistency (convergence to the true parameters in the large data limit) and asymptotic efficiency (achieves the Cramér-Rao Lower Bound, which is a lower bound on the variance of any unbiased estimators) - see Casella & Berger (2002) for an introduction. A key desideratum for spread MLE is to analyse how these properties are affected. In appendix(E) we demonstrate that spread MLE (for a certain family of spread noise) needs weaker sufficient conditions than MLE for both consistency and asymptotic efficiency. Furthermore, a sufficient condition for the existence of MLE is that the likelihood function is continuous over a compact parameter space $\Theta$. We provide an example in appendix(E.1) where the compactness requirement is violated, but spread MLE is still well defined.

## 5.2 APPLICATIONS

As an application to show the effectiveness of spread MLE, we use it to train implicit models

$$p_\theta(x) = \int \delta\left(x - g_\theta(z)\right) p(z)dz \tag{18}$$

where $\theta$ are the parameters of the encoder $g$. We show that, despite the likelihood not being defined (see also section(K) for a simple linear model example), we can nevertheless successfully train such models using modified EM/variational algorithms (Barber (2012)).

### 5.2.1 TRAINING IMPLICIT LINEAR MODELS: DETERMINISTIC ICA

ICA (Independent Components Analysis) corresponds to the model $p(x, z) = p(x|z) \prod_i p(z_i)$, where the independent components $z_i$ follow a non-Gaussian distribution. For Gaussian noise ICA an observation $x$ is assumed to be generated by the process $p(x|z) = \prod_j \mathcal{N}\left(x_j|g_j(z), \gamma^2\right)$ where $g_i(z)$ mixes the independent latent process $z$. In linear ICA, $g_j(z) = a_j^\mathsf{T} z$ where $a_j$ is the $j^{th}$ column on the mixing matrix $A$. For small observation noise $\gamma^2$, it is well known that the maximum likelihood EM algorithm to learn $A$ from observed data is ineffective (Bermond & Cardoso, 1999; Winther & Petersen, 2007). To see this, consider $X = Z$ (where $X$ and $Z$ are the dimension of the data and latents respectively) and invertible $A$, $x = Az$. At iteration $k$ the EM algorithm has an estimate $A_k$ of the mixing matrix. The M-step updates $A_k$ to

$$A_{k+1} = \mathbb{E}\left[xz^\mathsf{T}\right]\mathbb{E}\left[zz^\mathsf{T}\right]^{-1} \tag{19}$$

where, for zero observation noise ($\gamma = 0$),

$$\mathbb{E}\left[xz^\mathsf{T}\right] = \frac{1}{N}\sum_n x_n\left(A_k^{-1}x_n\right)^\mathsf{T} = \hat{S}A_k^{-\mathsf{T}}, \quad \mathbb{E}\left[zz^\mathsf{T}\right] = A_k^{-1}\hat{S}A_k^{-\mathsf{T}} \tag{20}$$

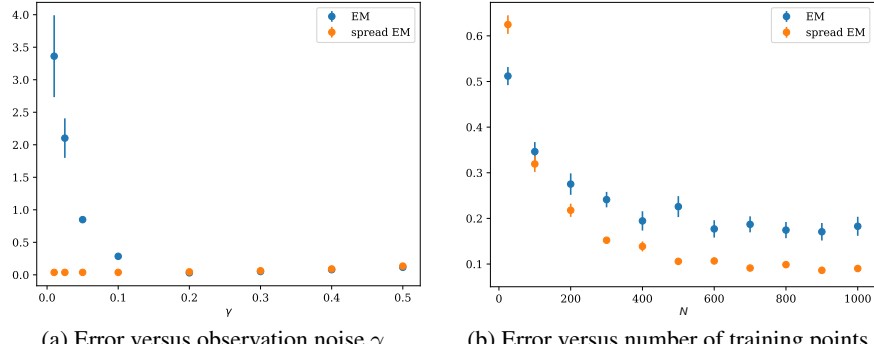

(a) Error versus observation noise $\gamma$.      (b) Error versus number of training points.

Figure 1: Relative error $|A_{ij}^{est} - A_{ij}^{true}|/|A_{ij}^{true}|$ versus observation noise (a) and number of training points (b). (a) For $X=20$ observations and $Z=10$ latent variables, we generate $N=20000$ datapoints from the model $x=Az$, for independent zero mean unit variance Laplace components on $z$. The elements of $A$ used to generate the data are uniform random $\pm 1$. We use $S_y=1$, $S_z=1000$ samples and 2000 EM iterations to estimate $A$. The error is averaged over all $i, j$ and 10 experiments. We also plot standard errors around the mean relative error. In blue we show the error in learning the underlying parameter using the standard EM algorithm. As expected, as $\gamma \to 0$, the error blows up as the EM algorithm 'freezes'. In orange we plot the error for EM using spread noise; no slowing down appears as the observation noise $\gamma$ decreases. In (b), apart from very small $N$, the error for the spread EM algorithm is lower than for the standard EM algorithm. Here $Z=5$, $X=10$, $S_y=1$, $S_z=1000$, $\gamma = 0.2$, with 500 EM updates used. Results are averaged over 50 runs of randomly drawn $A$.

and $\hat{S} \equiv \frac{1}{N} \sum_n x_n x_n^\mathsf{T}$ is the moment matrix of the data. Thus, $A_{k+1} = \hat{S} A_k^{-\mathsf{T}} \left( A_k^{-1} \hat{S} A_k^{-\mathsf{T}} \right)^{-1} = A_k$ and the algorithm 'freezes'. Similarly, for low noise $\gamma \ll 1$ progress critically slows down.

To deal with small noise and the limiting case of a deterministic model ($\gamma = 0$), we consider Gaussian spread noise $p(y|x) = \mathcal{N}\left(y|x, \sigma^2 I_X\right)$ to give

$$p(y, z) = \int p(y|x)p(x, z)dx = \prod_j \mathcal{N}\left(y_j|g_j(z), \left(\gamma^2 + \sigma^2\right) I_X\right) \prod_i p(z_i). \tag{21}$$

Using spread noise, the empirical distribution is replaced by the spreaded empirical distribution $\hat{p}(y) = \frac{1}{N} \sum_n \mathcal{N}\left(y|x^n, \sigma^2 I_X\right)$ The M-step has the same form as equation 19 but with modified statistics

$$\mathbb{E}\left[yz^\mathsf{T}\right] = \frac{1}{N} \sum_n \int \mathcal{N}\left(y|x^n, \sigma^2\right) p(z|y)yz^\mathsf{T}dzdy,$$

$$\mathbb{E}\left[zz^\mathsf{T}\right] = \frac{1}{N} \sum_n \int \mathcal{N}\left(y|x^n, \sigma^2\right) p(z|y)zz^\mathsf{T}dzdy. \tag{22}$$

The E-step optimally sets

$$p(z|y) = \frac{1}{Z_q(y)} \mathcal{N}\left(z|\mu(y), \Sigma\right) \prod_i p(z_i), \qquad Z_q(y) = \int \mathcal{N}\left(z|\mu(y), \Sigma\right) \prod_i p(z_i)dz \tag{23}$$

where $Z_q(y)$ is a normaliser and

$$\Sigma = \left(\gamma^2 + \sigma^2\right) \left(A^\mathsf{T}A\right)^{-1}, \qquad \mu(y) = \left(A^\mathsf{T}A\right)^{-1} Ay. \tag{24}$$

Since the posterior $p(z|y)$ peaks around $\mathcal{N}\left(z|\mu(y), \Sigma\right)$, we rewrite equation 22 as

$$\mathbb{E}\left[yz^\mathsf{T}\right] = \frac{1}{N} \sum_n \int \mathcal{N}\left(y|x^n, \sigma^2\right) \mathcal{N}\left(z|\mu(y), \Sigma\right) \frac{\prod_i p(z_i)}{Z_q(y)} yz^\mathsf{T}dzdy$$

and similarly for $\mathbb{E}\left[zz^\mathsf{T}\right]$. Writing the expectations with respect to $\mathcal{N}\left(z|\mu(y), \Sigma\right)$ allows for a simple but effective importance sampling approximation focused on regions of high probability.

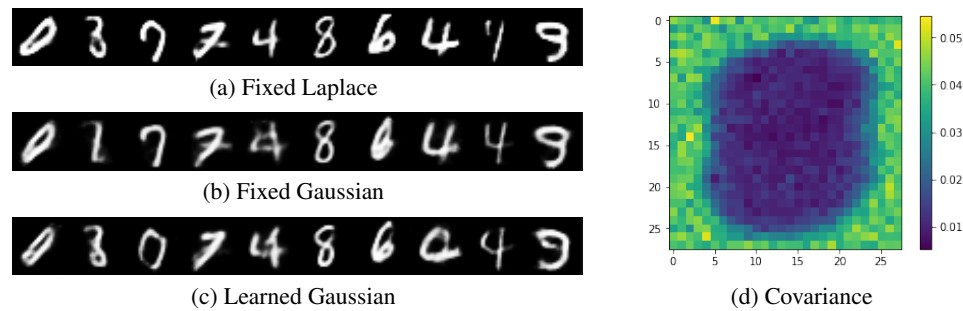

(a) Fixed Laplace

(b) Fixed Gaussian

(c) Learned Gaussian

(d) Covariance

Figure 2: Samples from a generative model (deterministic output) trained using $\delta$-VAE with (a) fixed Laplace covariance, (b) fixed Gaussian covariance and (c) learned Gaussian covariance. We first train with one epoch a standard VAE as initialization to all models, and keep latent code $z \sim \mathcal{N}\left(z|0, I_Z\right)$ fixed when sampling from these models, so we can more easily compare the sample quality. Figure (d) visualizes the absolute mean of the leading 20 eigenvectors of the learned covariance.

We implement this update by drawing $S_y$ samples from $\mathcal{N}\left(y|x_n, \sigma^2 I_X\right)$ and, for each $y$ sample, we draw $S_z$ samples from $\mathcal{N}\left(z|\mu(y), \Sigma\right)$. This scheme has the advantage over more standard variational approaches, see for example Winther & Petersen (2007), in that we obtain a consistent estimator of the M-step update for $A$. We show results for a toy experiment in figure(1), learning the underlying mixing matrix in a deterministic non-square setting. Note that standard algorithms such as FastICA (Hyvärinen, 1999) fail in this setting. The spread noise is set to $\sigma = \max(0.001, 2.5 *$ sqrt(mean($AA^\mathsf{T}$))). This modified EM algorithm thus learns a good approximation of the underlying $A$, with no critical slowing down. Other applications of EM that suffer from this slow down phenomenon, such as MLE for Slow Feature Analysis (Turner & Sahani (2007)), or other probabilistic matrix factorization algorithms (Barber (2012)), can also benefit from spread divergence.

### 5.2.2 TRAINING IMPLICIT NON-LINEAR MODELS: $\delta$-VAE

A standard way to train a deep generative model $p_\theta(x) = \int p_\theta(x|z)p(z)dz$ is to use maximum likelihood (minimizing $\mathrm{D}(\hat{p}(x)||p_\theta(x))$). The likelihood equation 18 is in general intractable and it is common to use the variational lower bound (see (Kingma & Welling, 2013)). However, for a deterministic observation model $p_\theta(x|z){=}\delta\left(x - g_\theta(z)\right)$ and $Z{<}X$, this generative model describes only a low dimensional manifold in the data space and the divergence $\mathrm{D}(\hat{p}(x)||p_\theta(x))$ is not well defined. Additionally the above bound is not well defined (due to log of a delta function) and the variational EM approach fails, as in the deterministic ICA setting. To address this, we instead minimize the spread divergence $\mathrm{KL}(\tilde{p}(y)||\tilde{p}_\theta(y))$. For Gaussian noise with fixed diagonal noise $p(y|x){=}\mathcal{N}\left(y|x, \sigma^2 I_X\right)$, we can write $\tilde{p}(y) = \frac{1}{N}\sum_{n=1}^N \mathcal{N}\left(y|x_n, \sigma^2 I_X\right)$ and

$$\tilde{p}_\theta(y) = \int p(y|x)p_\theta(x)dx = \int \mathcal{N}\left(y|g_\theta(z), \sigma^2 I_X\right) p(z)dz = \int p_\theta(y|z)p(z)dz. \quad (25)$$

We then minimize the divergence

$$\mathrm{KL}(\tilde{p}(y)||\tilde{p}_\theta(y)) = -\int \tilde{p}(y) \log \tilde{p}_\theta(y)dy + const. \quad (26)$$

Typically, the integral over $y$ is intractable, in which case we resort to a sampling estimation. Neglecting constants, the divergence estimator is $\frac{1}{NS}\sum_{n=1}^N \sum_{s=1}^S \log \tilde{p}_\theta(y_s^n)$, where $y_s^n$ is a spread noise sample from $p(y_n|x_n)$; for example $y_s^n \sim \mathcal{N}\left(y_s^n|x_n, \sigma^2 I_X\right)$. For non-linear $g$, the distribution $\tilde{p}_\theta(y)$ is usually intractable and we therefore use the variational lower bound

$$\log \tilde{p}_\theta(y) \geq \int q_\phi(z|y) \left(- \log q_\phi(z|y) + \log\left(p_\theta(y|z)p(z)\right)\right) dz. \quad (27)$$

The approach is a straightforward extension of the standard variational autoencoder and in appendix(G) we also derive a lower variance objective and detail how to learn the spread noise (also see appendix(F)). We dub this model and associated spread divergence training the '$\delta$-VAE'.

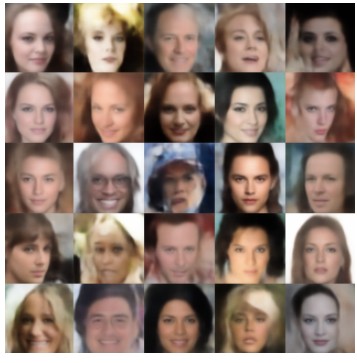 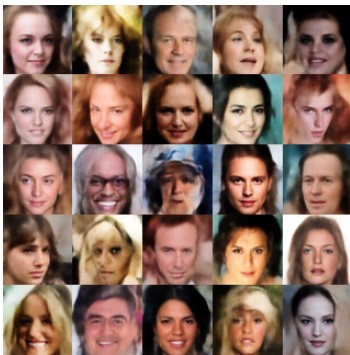

(a) $\delta$ Fixed spread noise  (b) $\delta$ Learned spread noise

Figure 3: Samples from a generative model with deterministic output trained using $\delta$-VAE with (a) fixed and (b) learned spread with injective function. We use a similar sampling strategy as in the MNIST experiment to facilitate sample comparison between the different models – see section(I).

| Encoder-Decoder Models | FID | GAN Models | FID |
|---|---|---|---|
| VAE | 63.0 | WGAN GP | 30.0 |
| $\delta$-VAE with fixed spread | **52.7** | BEGAN | 38.9 |
| $\delta$-VAE with learned spread | **46.5** | WGAN | 41.3 |
| | | DRAGAN | 42.3 |
| WAE-MMD | 55.0 | LSGAN | 53.9 |
| WAE-GAN | 42.0 | NS GAN | 55.0 |
| | | MM GAN | 65.6 |

Table 1: CelebA FID Scores. The $\delta$-VAE results are the average over 5 independent measurements. The scores of GAN-based models are based on a large-scale hyperparameter search and take the best FID obtained Lucic et al. (2018). The results of VAE and WAE-based model are from Tolstikhin et al. (2017).

**MNIST Experiment:** We trained a $\delta$-VAE on MNIST (LeCun et al. (2010)) with (i) fixed Laplace spread noise, equation 15, (ii) fixed Gaussian spread noise, equation 14 and (iii) Gaussian noise with learned covariance, section(4.1) with rank $R = 20$; see appendix(H) for details. Figures 2(a,b,c) show samples from $p_\theta(x)$ for these models; MNIST is sufficiently easy that it is hard to distinguish between the quality of the fixed and learned noise samples. However, qualitatively, the sharpness of the Laplace spread noise trained model is higher than for the Gaussian noise and motivates that the spread noise can affect the quality of the learned model. We speculate that Laplace noise improves image sharpness since the noise focuses attention on discriminating between points close to the data manifold (since the Laplace distribution is leptokurtic and has a higher probability of generating points close to the data manifold than the Gaussian distribution). Figure 2(d) visualizes the Gaussian learned covariance and shows that the learned noise is largely orthogonal to the data manifold.

**CelebA Experiment:** We trained a $\delta$-VAE on the CelebA dataset (Liu et al., 2015) with (i) fixed and (ii) learned spread with injective function, see appendix(I). We compared to results from a standard VAE with fixed Gaussian noise $p(x|z) = \mathcal{N}(x|g_\theta(z), 0.5I_X)$ Tolstikhin et al. (2017) . For (i) the fixed spread divergence uses Gaussian noise $\mathcal{N}(y|x, 0.25I_X)$. For (ii) we use Gaussian noise with learned injective function ResNet $f_\psi(\cdot) = I(\cdot) + g_\psi(\cdot)$; see appendix(I) for more details. Figure 3 shows samples from $\delta$-VAE trained using Gaussian spread divergence with both fixed and learned spread noise (with $g_\theta(z)$ initialised to the fixed-noise setting). It is notable how the 'sharpness' of the image samples substantially increases when learning the spread noise. Table 1 shows FID (Heusel et al. (2017)) score comparisons between different algorithms[3]. The $\delta$-VAE significantly improves on the standard VAE result; $\delta$-VAE with injective function learning also improves on the fixed-noise $\delta$-VAE. Indeed the injective $\delta$-VAE results are comparable to popular GAN and WAE models (Gulrajani et al. (2017); Berthelot et al. (2017); Arjovsky et al. (2017); Kodali et al. (2017); Mao et al. (2017); Fedus et al. (2017); Tolstikhin et al. (2017)). Whilst the $\delta$-VAE results are not fully state-of-the-art, we believe it is the first time that implicit models have been trained using a principled maximum likelihood based approach. Our expectation is that by increasing the complexity of the generative model $g_\theta$ and injective function $f_\psi$, or using different noise such as Laplace distribution, the results will become competitive with state-of-the-art GAN models.

---

[3]FID scores were computed using `github.com/bioinf-jku/TTUR` based on 10000 samples.

## 6 RELATED WORK

**MMD versus spread $f$-divergence:** In spite of the conditions required for defining the spread divergence being closely related to the kernel requirement of MMD (Gretton et al., 2012), we also show that MMD and spread Total Variation distance[4] can be written as different norms ($L_2$, $L_1$ respectively) of a common objective (see appendix(C)).

**Instance noise:** The instance noise trick to stabilize GAN training Roth et al. (2017); Sønderby et al. (2016) is a special case of spread divergence using fixed Gaussian noise. Whilst other similar tricks (for example Furmston & Barber (2009)) have been proposed previously, we believe that it is important to state the general utility of the spread noise approach.

**$\delta$-VAE versus WAE**: The Wasserstein auto-encoder Tolstikhin et al. (2017) is another implicit generative model that uses an encoder-decoder architecture. The difference is that $\delta$-VAE is based on KL divergence which is corresponding to MLE but WAE uses the Wasserstein distance.

**$\delta$-VAE versus denoising VAE**: The Denoising VAE Im et al. (2017) uses a VAE with noise added to the data only. In contrast, the $\delta$-VAE adds noise to both the data and model. Since the denoising VAE model only adds noise to the model, it cannot recover the true data distribution.

**MMD GAN with kernel learning**: The idea of learning a kernel to increase discrimination is also used in MMD GAN (Li et al. (2017)). Similar to ours, the kernel in MMD GAN is constructed by $\tilde{k} = k \circ f_\psi$ where $k$ is a fixed kernel and $f_\psi$ is a neural network. To ensure $M_{k \circ f_\psi}(p, q) = 0 \Leftrightarrow p = q$, this requires $f_\psi$ to be injective (Gretton et al. (2012)). However, in the MMD GAN framework, $f_\psi(x)$ usually maps $x$ to a lower dimension. This is crucial for MMD because the amount of data required to produce a reliable estimator grows with the data dimension (Ramdas et al. (2015)) and the computation cost of MMD scales quadratically with the amount of data. Whilst using a lower-dimensional mapping makes MMD more practical it also makes it difficult to construct an injective function $f$. For this reason, heuristics such as the auto-encoder regularizer (Li et al. (2017)) are considered. In contrast, for the $\delta$-VAE, the computational cost of estimating the divergence is linear in the number of datapoints. For this reason there is no need for $f_\psi$ to be a lower-dimensional mapping; guaranteeing that $f_\psi$ is injective is therefore relatively straightforward for the $\delta$-VAE.

**Flow-based generative models**: Invertible flow-based functions (Rezende & Mohamed (2015)) have been used to boost the representation power of generative models. Note our use of injective functions is quite distinct from the use of flow-based functions to boost generative model capacity. In our case, the injective function $f$ does not change the model – it only changes the divergence. For this reason, the spread divergence doesn't require the log determinant of the Jacobian (which is required in Rezende & Mohamed (2015); Behrmann et al. (2018)) meaning that more general invertible functions can be used to boost the discriminatory power of a spread divergence.

## 7 SUMMARY

We described how to define a divergence even when two distributions do not have the same support. Previous approaches (Furmston & Barber, 2009; Sønderby et al., 2016) can be seen as special cases. We showed that defining divergences this way enables us to train deterministic generative models using standard likelihood based approaches. In principle, we can learn the underlying true data generating process by the use of any valid spread divergence. In practice, however, the quality of the learned model can depend strongly on the choice of spread noise. We therefore investigated learning spread noise to maximally discriminate two distributions. We found the resulting training approach stable and that it can significantly improve the image generation results. Whilst state-of-the-art image generation is not the focus of this work, we obtained promising results. We also discussed the conditions under which spread MLE is consistent and asymptotically efficient, some of which are weaker than the equivalent MLE conditions. Perhaps the most appealing aspect of the spread noise is that is enables one to re-use standard machine learning approaches in statistics such as maximum likelihood to train models that would be otherwise unsuited to standard statistical training approaches.

---

[4]Total Variation distance between $p(x)$ and $q(x)$ is defined (up to a constant scale) as $TV(p(x)||q(x)) = \int |p(x) - q(x)| dx$, it belongs to $f$-divergence family (see Liese & Vajda (2006)).

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

## A  PROOF OF THEOREM A

**Theorem 1.** *Consider distributions $p$, $q$, and $L^1$ integrable function $K$ and Fourier Transform $\mathcal{F}\{K\}$. Let $\mathcal{F}\{K\} \neq 0$ or $\mathcal{F}\{K\} = 0$ on at most a countable set. Then*

$$\mathcal{F}\{K\}\mathcal{F}\{p\} = \mathcal{F}\{K\}\mathcal{F}\{q\} \Rightarrow \mathcal{F}\{p\} = \mathcal{F}\{q\}. \tag{28}$$

*Proof.* When $\mathcal{F}\{K\} \neq 0$, $\mathcal{F}\{K\}\mathcal{F}\{p\} = \mathcal{F}\{K\}\mathcal{F}\{q\} \Rightarrow \mathcal{F}\{p\} = \mathcal{F}\{q\}$ is trivial. We first show that the Fourier transform of an $L^1$ function is continuous on $\mathbb{R}^d$. When $\mathcal{F}\{K\} = 0$ on at most a countable set, we then show that $\mathcal{F}\{q\}$ and $\mathcal{F}\{p\}$ cannot be different at a set of countable points.

Since any distribution $q$ is in $L^1$, we can write

$$\lim_{\epsilon \to 0} |\mathcal{F}\{q\}(w + \epsilon) - \mathcal{F}\{q\}(w)| = \lim_{\epsilon \to 0} \left| \int q(x) \left( e^{-2\pi i x(w+\epsilon)} - e^{-2\pi i x w} \right) dx \right|$$

$$\leq \lim_{\epsilon \to 0} \int |q(x)| \left| e^{-2\pi i x \epsilon} - 1 \right| dx$$

$$= \int \lim_{\epsilon \to 0} |q(x)| \left| e^{-2\pi i x \epsilon} - 1 \right| dx \text{ (Dominated Convergence Theorem)}$$

$$= 0$$

So $\mathcal{F}\{q\}$ is (uniformly) continuous. The same argument applies to show that $\mathcal{F}\{p\}$ is uniformly continuous.

Since $\mathcal{F}\{K\} = 0$ on at most a countable set $\mathcal{C}$, we assume there is a point $w_0 \in \mathcal{C}$ where $\mathcal{F}\{q\}(w_0) \neq \mathcal{F}\{p\}(w_0)$. Without loss of generality, we assume $\mathcal{F}\{q\}(w_0) - \mathcal{F}\{p\}(w_0) = \theta > 0$. For points $w_0 + h$ that are not in $\mathcal{C}$, $\mathcal{F}\{K\}(w_0 + h) \neq 0$ and it follows therefore that $\mathcal{F}\{K\}\mathcal{F}\{p\} = \mathcal{F}\{K\}\mathcal{F}\{q\}$ implies $\mathcal{F}\{q\}(w_0 + h) - \mathcal{F}\{p\}(w_0 + h) = 0$. By continuity of $\mathcal{F}\{p\}$ and $\mathcal{F}\{q\}$, we have $\mathcal{F}\{q\}(w_0 + h) - \mathcal{F}\{p\}(x_0 + h) \to 0$ when $h \to 0$, which leads to a contradiction ($\theta$ cannot be zero). $\square$

## B  SPREAD NOISE MAKES DISTRIBUTIONS MORE SIMILAR

The data processing inequality for $f$-divergences (see for example Gerchinovitz et al. (2018)) states that $\mathrm{D}_f(\tilde{p}(y)||\tilde{q}(y)) \leq \mathrm{D}_f(p(x)||q(x))$. For completeness, we provide here an elementary proof of this result. We consider the following joint distributions

$$q(y, x) = p(y|x)q(x), \qquad p(y, x) = p(y|x)p(x) \tag{29}$$

whose marginals are the spreaded distributions

$$\tilde{p}(y) = \int_x p(y|x)p(x), \qquad \tilde{q}(y) = \int_x p(y|x)q(x) \tag{30}$$

The divergence between the two joint distributions is

$$\mathrm{D}_f(p(y, x)||q(y, x)) = \int_{x,y} q(y, x)f\left(\frac{p(y|x)p(x)}{p(y|x)q(x)}\right) = \mathrm{D}_f(p(x)||q(x)) \tag{31}$$

The $f$-divergence between two marginal distributions is no larger than the $f$-divergence between the joint (see also Zhang et al. (2018)). To see this, consider

$$\mathrm{D}_f(p(u, v)||q(u, v)) = \int q(u) \int q(v|u)f\left(\frac{p(u, v)}{q(u, v)}\right) dy du$$

$$\geq \int q(u)f\left(\int q(v|u)\frac{p(u, v)}{q(v|u)q(u)}dv\right) du$$

$$= \int q(u)f\left(\frac{p(u)}{q(u)}\right) du = \mathrm{D}_f(p(u)||q(u))$$

Hence,

$$\mathrm{D}_f(\tilde{p}(y)||\tilde{q}(y)) \leq \mathrm{D}_f(p(y, x)||q(y, x)) = \mathrm{D}_f(p(x)||q(x)) \tag{32}$$

Intuitively, spreading two distributions increases their overlap, reducing the divergence. When $p$ and $q$ do not have the same support, $\mathrm{D}_f(q(x)||p(x))$ can be infinite or not well-defined.

## C    Relation to MMD

Spread divergence can be generally constructed from $f$-divergence, we show how to build a connection to maximum mean discrepancy (Gretton et al. (2012)) by using the spread total variation distance:

For a translation invariant kernel $k(\cdot, x)$, the MMD is (we use $\gamma_k$ to denote the MMD distance with kernel $k$)

$$\gamma_k(p(x)||q(x)) = \left\| \int k(\cdot, x)p(x)dx - \int k(\cdot, x)q(x)dx \right\|_{\mathcal{H}}$$

Suppose both $k$ and it's square root of Fourier transform $\sqrt{\hat{k}}$ (ˆrepresents the Fourier transform) are absolutely integratable, we can also rewrite the MMD distance the as following (see Sriperumbudur et al. (2010) for a derivation):

$$\gamma_k(p||q) = (2\pi)^{-d/4}||\Phi * p - \Phi * q||_{L_2},$$

where $\Phi := (\sqrt{\hat{k}})^{\vee}$ ($\vee$ represents the inverse Fourier transform).

We can further define $\tilde{\Phi} = Z^{-1}\Phi$, where $Z = \int \Phi(x)dx$. Therefore, $\tilde{\Phi}$ is a probability density function and the MMD can be written as

$$\gamma_k(p||q) = (2\pi)^{-d/4}Z||\tilde{\Phi} * p - \tilde{\Phi} * q||_{L_2}$$
$$\propto ||\tilde{\Phi} * p - \tilde{\Phi} * q||_{L_2}.$$

Total variation distance between $p(x)$ and $q(x)$ is defined as

$$TV(p||q) = ||p - q||_{L_1}.$$

In the spread total variation distance, we can define spread as the noise distribution $k = \tilde{\Phi}$, thus

$$\widetilde{TV}(p||q) = ||\tilde{\Phi} * p - \tilde{\Phi} * q||_{L_1}$$

So MMD and spread Total Variation distance can be written as different norms of a common objective.

## D    Spread divergence between two delta distributions

Let $p_0(x) = \delta(x - \mu_0)$, $p_q(x) = \delta(x - \mu_1)$, assume Gaussian spread noise $p(y|x) = \mathcal{N}\left(y|x, \sigma^2\right)$ and $\sigma^2 = 0.5$, so

$$\widetilde{KL}(p_0(x)||p_1(x)) = KL(\tilde{p}_0(y)||\tilde{p}_1(y))$$
$$= KL\left(\int_x p(y|x)p_0(x)|| \int_x p(y|x)p_1(x)\right)$$
$$= KL\left(\mathcal{N}\left(y|\mu_1, \sigma^2\right)||\mathcal{N}\left(y|\mu_2, \sigma^2\right)\right)$$
$$= \log\frac{\sigma^2}{\sigma^2} + \frac{\sigma^2 + (\mu_1 - \mu_2)^2}{2\sigma^2} - \frac{1}{2}$$
$$= (\mu_1 - \mu_2)^2$$

## E    Statistical Properties of Maximum Likelihood Estimator

### E.1    Existence of Spread MLE

In some situations there may not exist a Maximum Likelihood Estimator (MLE) for $p(x|\theta)$, but there is a MLE for the spread model $p(y|\theta) = \int p(y|x)p(x|\theta)dx$. For example, suppose that $X \sim N(\mu, \sigma^2)$ $(\mu, 0 < \sigma^2 < \infty)$. So $\theta = (\mu, \sigma^2) \in \mathbb{R} \times \mathbb{R}^+$. Assume we only have one data point $x$. Then the log-likelihood function is $L(x; \theta) \propto -\log\sigma - \frac{1}{2\sigma^2}(x - \mu)^2$. Maximising with respect to $\mu$, we have $\mu = x$ and the log-likelihood becomes unbounded as $\sigma^2 \to 0$. In this sense, the MLE for $(\mu, \sigma^2)$ does not exist.

On the other hand, we can check whether the MLE for $p(y|\theta)$ exists. We assume Gaussian spread noise with fixed variance $\sigma_f^2$. Since we only have one data point $x$, the spread data distribution becomes $p(y|x) = \mathcal{N}\left(y|x, \sigma_f^2\right)$, and the model is $p(y|\theta) = \mathcal{N}\left(y|\mu, \sigma^2 + \sigma_f^2\right)$. We can sample $N$ points from the spread model, so the spread log likelihood function is (neglecting constants) $L(y_1, \ldots, y_N; \theta) = -\frac{N}{2}\log(\sigma^2 + \sigma_f^2) - \frac{1}{2(\sigma^2 + \sigma_f^2)}\sum_{i=1}^{N}(y_i - \mu)^2$. The MLE solution for $\mu$ is $\mu = \frac{1}{N}\sum_{i=1}^{N} y_i$; the MLE solution for $\sigma^2$ is $\sigma^2 = \frac{1}{N}\sum_i (y_i - \mu)^2 - \sigma_f^2$, which has bounded spread likelihood value. Note that in the limit of a large number of spread samples $N \to \infty$, the MLE $\sigma^2 = \frac{1}{N}(y_i - \mu)^2 \to \sigma_f^2$ tends to 0. Throughout, however, the (scaled by $N$) log likelihood remains bounded.

## E.2 CONSISTENCY

Consistency of an estimator is an important property that guarantees the validity of the resulting estimate at convergence as the number of data points tends to infinity. In what follows, we refer to sufficient conditions for a consistent MLE estimator, before addressing the question of whether using spread MLE is also consistent and under what conditions.

### E.2.1 CONSISTENCY FOR MLE

Sufficient conditions for the MLE being consistent and converging to the *global* maximum are given in Wald (1949). However, they are usually difficult to check even for some standard distributions. The sufficient conditions for MLE being consistent and converging to a *local* maxima are given in Cramér (1999) and are more straight forward to check:

    C1. (Identifiable): $p(x|\theta_1) = p(x|\theta_2) \to \theta_1 = \theta_2$.

    C2. The parameter space $\Theta$ is an open interval $(\underline{\theta}, \bar{\theta})$, $\Theta : -\infty \leq \underline{\theta} < \theta < \bar{\theta} \leq \infty$.

    C3. $p(x|\theta)$ is continuous in $\theta$ and differentiable with respect to $\theta$ for all x.

    C4. The set $A = \{x : p_\theta(x) > 0\}$ is independent of $\theta$.

Let $X_1, X_2, \ldots$ be $i.i.d$ with density $p(x|\theta_0)$ ($\theta \in \Theta$) satisfying conditions C1–C4, then there exists a sequence $\hat{\theta}_n = \hat{\theta}_n(X_1, ..., X_n)$ of local maxima of the likelihood function $L(\theta_0) = \prod_{i=1}^{n} p(x_i|\theta_0)$ which is consistent:

$$\hat{\theta} \xrightarrow{p} \theta_0 \quad \text{for all } \theta \in \Theta$$

The proof can be found in Lehmann (2004) or Cramér (1999).

### E.2.2 CONSISTENCY OF SPREAD MLE

We provide the necessary conditions for Spread MLE being consistent.

    C1. (Identifiable): $p(x|\theta)$ is identifiable. From section(3) it follows immediately that $p(y|\theta_1) = p(y|\theta_2) \to p(x|\theta_1) = p(x|\theta_2) \to \theta_1 = \theta_2$, where the final implication follows from the assumption that $p(x|\theta)$ is identifiable. Hence if $p(x|\theta)$ is identifiable, so is $p(y|\theta)$.

    C2. The parameter space $\Theta$ is an open interval $(\underline{\theta}, \bar{\theta})$, $\Theta : -\infty \leq \underline{\theta} < \theta < \bar{\theta} \leq \infty$. This condition is unchanged for $p(y|\theta)$.

    C3. On $p(y|\theta)$, we require the same condition on $p(x|\theta)$ as in MLE; $p(y|\theta)$ is continuous in $\theta$ and differentiable with respect to $\theta$ for all $y$.

    C4. For spread noise $p(y|x)$ who has full support on $\mathbb{R}^d$ (for example Gaussian noise), $p(y|\theta)$ is greater than zero everywhere and hence the original condition C4 is automatically guaranteed.

The conditions that guarantee consistency for spread MLE are weaker for the spread model $p(y|\theta)$ than for the standard model $p(x|\theta)$, since C4 is automatically satisfied. Ferguson (1982) gives an example for which MLE exists but is not consistent by violating condition C4, whereas spread MLE can be used to obtain a consistent estimator.

### E.3    ASYMPTOTIC EFFICIENCY

A key desirable property of any estimator is that it is efficient. The Cramer-Rao bound places a lower bound on the variance of any unbiased estimator and an efficient estimator much reach this minimal value in the limit of a large amount of data. Under certain conditions (see below) the Maximum Likelihood Estimator attains this minimal variance value meaning that there is no better estimator possible than maximum likelihood (in the limit of a large amount of data). This is one of the reasons that the maximum likelihood is a cherished criterion.

#### E.3.1    ASYMPTOTIC EFFICIENCY FOR MLE

Building upon conditions C1-C4, additional conditions on $p(x|\theta)$ are required to show MLE is asymptotical efficient:

C5.  For all $x$ in its support, the density $p_\theta(x)$ is three times differentiable with respect to $\theta$ and the third derivative is continuous.

C6.  The derivatives of the integral $\int p_\theta(x)dx$ respect to $\theta$ can be obtained by differentiating under the integral sign, that is: $\nabla_\theta \int p_\theta(x)dx = \int \partial_\theta p_\theta(x)dx$.

C7.  There exists a positive number $c(\theta_0)$ and a function $M_{\theta_0}(x)$ such that

$$|\frac{\partial^3}{\partial\theta^3} \log p_\theta(x)| \leq M_{\theta_0}(x) \quad \text{for all } x \in A, |\theta - \theta_0| < c(\theta_0)$$

where $A$ is the support set of $x$ and $\mathbb{E}_{\theta_0}[M_{\theta_0}(x)] < \infty$.

Let $X_1, ..., X_n$ be $i.i.d$ with density $p_\theta(x)$ and satisfy conditions C1-C7, then any consistent sequence $\hat{\theta} = \hat{\theta}_n(X_1, ..., X_n)$ of roots of the likelihood equation satisfies

$$\sqrt{n}(\hat{\theta} - \theta_0) \xrightarrow{d} N(0, F(\theta_0)^{-1}),$$

where $F^{-1}(\theta_0)$ is the inverse of Fisher information matrix (also called Cramér-Rao Lower Bound, which is a lower bound on variance of any unbiased estimators ). The conditions and proof can be found in Lehmann (2004).

#### E.3.2    ASYMPTOTIC EFFICIENCY FOR MLE

As with MLE above, we require further conditions on $p(y|\theta)$ for ensuring spread MLE is asymptotically efficient:

C5.  On $p(y|\theta)$, we require the same condition as applied to $p(x|\theta)$ in the MLE case; for all $y$ in its support, the density $p_\theta(y)$ is three times differentiable with respect to $\theta$ and the third derivative is continuous.

C6.  For spread noise $p(y|x)$, which has full support on $\mathbb{R}^d$ (for example Gaussian noise), the support of $y$ is independent of $\theta$. Leibniz's rule[5] allows us to differentiate under the integral: $\nabla_\theta \int p_\theta(y)dy = \int \partial_\theta p_\theta(y)dy$, so this condition is automatically satisfied.

C7.  On $p(y|\theta)$, we require the same condition as applied to $p(x|\theta)$ in the MLE case; There exist positive number $c(\theta_0)$ and a function $M_{\theta_0}(y)$ such that

$$|\frac{\partial^3}{\partial\theta^3} \log p_\theta(y)| \leq M_{\theta_0}(y) \quad \text{for all } y \in A, |\theta - \theta_0| < c(\theta_0)$$

where $A$ is the support set of $y$ and $\mathbb{E}_{\theta_0}[M_{\theta_0}(y)] < \infty$.

Thus the conditions that guarantee asymptotically efficient for the spread model $p(y|\theta)$ are weaker than for the standard model $p(x|\theta)$, since C4 and C6 are automatically satisfied.

---

[5]Leibniz's rule tells us: $\frac{d}{d\theta} \int_{a(\theta)}^{b(\theta)} p(x,\theta)dx = \int_{a(\theta)}^{b(\theta)} \partial_\theta p(x,\theta)dx + p(b(\theta),\theta)\frac{d}{d\theta}b(\theta) - p(a(\theta),\theta)\frac{d}{d\theta}a(\theta)$, so if $a(\theta)$ and $b(\theta)$ are independent of $\theta$, then $\frac{d}{d\theta} \int_a^b p(x,\theta)dx = \int_a^b \partial_\theta p(x,\theta)dx$.

## F  PERTURBATION APPROXIMATION OF GAUSSIAN SPREAD

Herein, in the fixed noise setting, we derive a perturbation based approximation to the spread noise, which in princple can lead to a lower variance estimator. We can write a function $f$ with perturbed input and integrated over noise as

$$\mathbb{E}_{p(\xi)}\left[f(x+\xi)\right],\tag{33}$$

where $p(\xi) = \mathcal{N}(\xi|0,\Sigma)$. Taylor expanding around $\xi = 0$, we have

$$\mathbb{E}_{p(\xi)}\left[f(x+\xi)\right] \approx \mathbb{E}_{p(\xi)}\left[f(x) + \xi^\mathsf{T}\nabla f(x) + \frac{1}{2}\xi^\mathsf{T}\nabla^2 f(x)\xi + \mathcal{O}(\xi^3)\right]\tag{34}$$

$$\approx f(x) + \frac{1}{2}\mathbb{E}_{p(\xi)}\left[\xi^\mathsf{T}\mathbf{H}\xi\right]\tag{35}$$

$$= f(x) + \frac{1}{2}\mathrm{Tr}\left(\mathbb{E}_{p(\xi)}\left[\xi\xi^T\right]\mathbf{H}\right)\tag{36}$$

$$= f(x) + \frac{1}{2}\mathrm{Tr}\left(\Sigma\mathbf{H}\right)\tag{37}$$

Where $\mathbf{H}$ is the Hessian matrix $\mathbf{H}_{i,j} = \frac{\partial^2 f}{\partial x_i \partial x_j}$. When $x$'s dimension is small, we can use equation 37 to explicitly calculate the trace. When the dimension of $x$ is large, we can form a Monte Carlo estimation of equation 35. To do this we first sample $\xi \sim p(\xi)$ and then calculate the Hessian-vector product $\mathbf{H}\xi$. This can be efficiently calculated by AutoDiff backward mode (Schraudolph (2002); Pearlmutter (1994)), without storing the Hessian matrix in memory.

For example, in $\delta$-VAE with fixed Gaussian noise, according to the bound equation 56 (ignoring the constant):

$$\int \mathcal{N}\left(y|x,\sigma^2 I_X\right)\log \tilde{p}_\theta(y) \geq \mathbb{E}_{\mathcal{N}(\epsilon_x|0,\sigma^2 I_X)}\left[H(\Sigma_\phi(x+\sigma\epsilon_x))\right]\tag{38}$$

$$+\mathbb{E}_{\mathcal{N}(\epsilon_x|0,\sigma^2 I_X)}\left[\mathbb{E}_{\mathcal{N}(\epsilon|0,I)}\left[\log p(x=g_\theta(\mu_\phi(x+\sigma\epsilon_x)+C_\phi(x)\epsilon)) + \log p(z=\mu_\phi(x+\sigma\epsilon_x)+C_\phi(x)\epsilon)\right]\right]\tag{39}$$

$$\approx \underbrace{H(\Sigma_\phi(x)) + \mathbb{E}_{\mathcal{N}(\epsilon|0,I)}\left[\log p(x=g_\theta(\mu_\phi(x)+C_\phi(x)\epsilon)) + \log p(z=\mu_\phi(x)+C_\phi(x)\epsilon)\right]}_{f(x)} + \frac{\sigma^2}{2}\mathrm{Tr}\left(\mathbf{H}\right)\tag{40}$$

Where $\mathbf{H}$ is the Hessian matrix $\mathbf{H}_{i,j} = \frac{\partial^2 f(x)}{\partial x_i \partial x_j}$ and $C_\phi(x)$ is the Cholesky decomposition of $\Sigma_\phi(x)$.

## G  SPREAD DIVERGENCE FOR DETERMINISTIC DEEP GENERATIVE MODELS

Instead of minimising the likelihood, we train an implicit generative model by minimising the spread divergence

$$\min_\theta \mathrm{KL}(\tilde{p}(y)||\tilde{p}_\theta(y))\tag{41}$$

For Gaussian noise with fixed diagonal noise $p(y|x) = N(y|x,\sigma^2 I_X)$, we can write

$$\tilde{p}(y) = \frac{1}{N}\sum_{n=1}^N \mathcal{N}\left(y|x_n,\sigma^2 I_X\right)\tag{42}$$

and

$$\tilde{p}_\theta(y) = \int p(y|x)p_\theta(x)dx = \int \mathcal{N}\left(y|g_\theta(z),\sigma^2 I_X\right)p(z)dz = \int p_\theta(y|z)p(z)dz\tag{43}$$

For the spread divergence with learned covariance Gaussian noise which is discussed in section(4.1), we can write

$$p_\psi(y|x) = \mathcal{N}(y|x,\Sigma_\psi), \qquad \tilde{p}(y) = \frac{1}{N}\sum_{n=1}^N \mathcal{N}\left(y|x_n,\Sigma_\psi\right)\tag{44}$$

and spread divergence with learned injective function as discussed in section(4.2)

$$p_\psi(y|x) = \mathcal{N}(y|f_\psi(x), \sigma^2 I_X), \qquad \tilde{p}(y) = \frac{1}{N} \sum_{n=1}^{N} \mathcal{N}(y|f_\psi(x), \sigma^2 I_X) \qquad (45)$$

According to our general theory,

$$\min_\theta \mathrm{KL}(\tilde{p}(y)||\tilde{p}_\theta(y)) = 0 \qquad \Leftrightarrow \qquad p(x) = p_\theta(x) \qquad (46)$$

Here

$$\mathrm{KL}(\tilde{p}(y)||\tilde{p}_\theta(y)) = -\frac{1}{N} \sum_{n=1}^{N} \int \tilde{p}(y) \log \tilde{p}_\theta(y) dy + const. \qquad (47)$$

Typically, the integral over $y$ will be intractable and we resort to an unbiased sampled estimate (though see below for Gaussian $q$). Neglecting constants, the KL divergence estimator is

$$\frac{1}{NS} \sum_{n=1}^{N} \sum_{s=1}^{S} \log \tilde{p}_\theta(y_s^n) \qquad (48)$$

where $y_s^n$ is a perturbed version of $x_n$. For example $y_s^n \sim \mathcal{N}\left(y_s^n|x_n, \sigma^2 I_X\right)$ for fixed Gaussian noise case and other cases are similar. In most cases of interest, with non-linear $g$, the distribution $\tilde{p}_\theta(y)$ is intractable. We therefore use the variational lower bound

$$\log \tilde{p}_\theta(y) \geq \int q_\phi(z|y) \left( -\log q_\phi(z|y) + \log \left( p_\theta(y \mid z) p(z) \right) \right) dz \qquad (49)$$

Parameterising the variational distribution as a Gaussian,

$$q_\phi(z|y) = \mathcal{N}\left(z|\mu_\phi(y), \Sigma_\phi(y)\right) \qquad (50)$$

then we can reparameterise and write

$$\log \tilde{p}_\theta(y) \geq H(\Sigma_\phi(y)) + \mathbb{E}_{\mathcal{N}(\epsilon|0,I)} \left[ \log \left( p_\theta(y|z = \mu_\phi + C_\phi(y)\epsilon) p(z = \mu_\phi(y) + C_\phi(y)\epsilon) \right) \right] \qquad (51)$$

where $H(\Sigma_\phi(y))$ is the entropy of a Gaussian with covariance $\Sigma_\phi(y)$. For fixed covariance Gaussian spread noise in $D$ dimensions, this is

$$\log \tilde{p}_\theta(y) \geq H(\Sigma_\phi(y)) + \mathbb{E}_{\mathcal{N}(\epsilon|0,I)} \left[ -\frac{1}{(2\sigma^2)^{D/2}} \left( y - g_\theta \left( \mu_\phi(y) + C_\phi(y)\epsilon \right) \right)^2 + \log p(z = \mu_\phi(y) + C_\phi(y)\epsilon) \right] + const. \qquad (52)$$

where $C_\phi(y)$ is the Cholesky decomposition of $\Sigma_\phi(y)$.

We can integrate equation 52 over $y$ to give the bound

$$\int \mathcal{N}\left(y|x, \sigma^2 I_X\right) \log \tilde{p}_\theta(y) \geq \mathbb{E}_{\mathcal{N}(y|x, \sigma^2 I_X)} \left[ H(\Sigma_\phi(y)) + \mathbb{E}_{\mathcal{N}(\epsilon|0,I)} \left[ \log p(z = \mu_\phi(y) + C_\phi(y)\epsilon) \right] \right] \qquad (53)$$

$$-\frac{1}{(2\sigma^2)^{D/2}} \mathbb{E}_{\mathcal{N}(\epsilon|0,I)} \left[ \mathbb{E}_{\mathcal{N}(y|x, \sigma^2 I_X)} \left[ \left( y - f_\psi(g_\theta \left( \mu_\phi(y) + C_\phi(y)\epsilon \right) ) \right)^2 \right] \right] + const. \qquad (54)$$

where

$$\mathbb{E}_{\mathcal{N}(y|x, \sigma^2 I_X)} \left[ \left( y - g_\theta \left( \mu_\phi(y) + C_\phi(y)\epsilon \right) \right)^2 \right]$$

$$= \sigma^2 - 2\mathbb{E}_{\mathcal{N}(\epsilon_x|0, I_X)} \left[ \epsilon_x g_\theta(\mu_\phi(x + \sigma \epsilon_x) + C_\phi(y)\epsilon) \right] + \mathbb{E}_{\mathcal{N}(\epsilon_x|0, I_X)} \left[ \left( x - g_\theta(\mu_\phi \left( x + \sigma \epsilon_x \right) + C_\phi(y)\epsilon) \right)^2 \right] \qquad (55)$$

We notice that the second term is zero, so the final bound for the fixed Gaussian spread KL divergence is (ignoring the constant)

$$\int \mathcal{N}\left(y|x, \sigma^2 I_X\right) \log \tilde{p}_\theta(y) \geq \mathbb{E}_{\mathcal{N}(y|x, \sigma^2 I_X)} \left[ H(\Sigma_\phi(y)) + \mathbb{E}_{\mathcal{N}(\epsilon|0,I)} \left[ \log p(z = \mu_\phi(y) + C_\phi(y)\epsilon) \right] \right]$$

$$-\frac{1}{(2\sigma^2)^{D/2}} \mathbb{E}_{\mathcal{N}(\epsilon_x|0, I_X)} \left[ \mathbb{E}_{\mathcal{N}(\epsilon|0,I)} \left[ \left( x - g_\theta(\mu_\phi \left( x + \sigma \epsilon_x \right) + C_\phi \epsilon) \right)^2 \right] \right] \qquad (56)$$

By analogy, for spread KL divergence with learned variance, the bound is (ignoring the constant)

$$\int \mathcal{N}\left(y|x, \Sigma_\psi\right) \log \tilde{p}_\theta(y) \geq \mathbb{E}_{\mathcal{N}(y|x,\Sigma_\psi)}\left[H(\Sigma_\phi(y)) + \mathbb{E}_{\mathcal{N}(\epsilon|0,I)}\left[\log p(z = \mu_\phi(y) + C_\phi(y)\epsilon)\right]\right]$$

$$- \mathbb{E}_{\mathcal{N}(\epsilon_x|0,\Sigma_\psi)}\left[\mathbb{E}_{\mathcal{N}(\epsilon|0,I)}\left[(x - g_\theta(\mu_\phi\left(x + S_\psi\epsilon_x\right) + C_\phi(y)\epsilon))^T \Sigma_\psi^{-1}\left(x - g_\theta(\mu_\phi\left(x + S_\psi\epsilon_x\right) + C_\phi(y)\epsilon)\right)\right]\right]$$

$$(57)$$

Where $S_\psi$ is the cholesky decomposition of $\Sigma_\psi$. For specific covariance structure introduced in section section(4.1), efficient methods for sampling, matrix inverting and log determinant calculation are available, see appendix(J).

For spread KL divergence with learned injective function, the bound is (ignoring the constant)

$$\int \mathcal{N}\left(y|f_\psi(x), \sigma^2 I_X\right) \log \tilde{p}_\theta(y) \geq \mathbb{E}_{\mathcal{N}(y|x,\sigma^2 I_X)}\left[H(\Sigma_\phi(y)) + \mathbb{E}_{\mathcal{N}(\epsilon|0,I)}\left[\log p(z = \mu_\phi(y) + C_\phi(y)\epsilon)\right]\right]$$

$$- \frac{1}{(2\sigma^2)^{D/2}}\mathbb{E}_{\mathcal{N}(\epsilon_x|0,I_X)}\left[\mathbb{E}_{\mathcal{N}(\epsilon|0,I)}\left[(f_\psi(x) - f_\psi(g_\theta(\mu_\phi\left(f_\psi(x) + \sigma\epsilon_x\right) + C_\phi(y)\epsilon)))^2\right]\right] \quad (58)$$

The overall procedure is therefore a straightforward modification of the standard VAE method Kingma & Welling (2013) with additional learning the spread to maximize the divergence:

1. Choose a noise distribution $p(y|x)$
2. Choose a tractable family for the variational distribution, for example $q_\phi(z|y) = \mathcal{N}\left(z|\mu_\phi(y), \Sigma_\phi(y)\right)$ and initialise $\phi$.
3. We then sample a $y_n$ for each datapoint (if we're using $S = 1$ samples)
4. If learning the spread noise:
   (a) Draw samples $\epsilon$ to estimate $-\log \tilde{p}_\theta(y_n)$ according to the corresponding bound.
   (b) Do a gradient ascent step in $\psi$.
5. Draw samples $\epsilon$ to estimate $\log \tilde{p}_\theta(y_n)$ according to the corresponding bound.
6. Do a gradient ascent step in $(\theta, \phi)$.
7. Go to 3 and repeat until convergence.

## H   MNIST EXPERIMENT

We first scaled the MNIST data to lie in $[0, 1]$. We use Laplace spread noise $\sigma = 0.3$ and Gaussian spread noise $\sigma = 0.3$ for the $\delta$VAE. Both encoder and decoder contains 3 feed-forward layers, each layer with 400 units and ReLu activation function. The latent dimension is $Z = 64$. The variational inference network $q_\phi(z|y) = \mathcal{N}\left(z|\mu_\phi(y), \sigma_\phi^2 I_Z\right)$ has a similar structure for the mean network $\mu_\phi(y)$. For fixed spread $\delta$-VAE , learning was done using the Adam Kingma & Ba (2014) optimizer with learning rate $5e^{-4}$ for 200 epochs. For $\delta$-VAE with learned spread (learned covariance), we additioanly train the covariance for 2 epochs using Adam optimizer with learning rate $5e^{-5}$ after everytime we train the model for 10 epochs.

## I   CELEBA EXPERIMENT

We pre-processed CelebA images by first taking 140x140 centre crops and then resizing to 64x64. Pixel values were then rescaled to lie in $[0, 1]$. For the learned spread we use Gaussian noise with learned injective function ResNet $f_\psi(\cdot) = I(\cdot) + g_\psi(\cdot)$, where $g_\psi(\cdot)$ is a one layer convolutional neural net with kernel size $3 \times 3$ and stride 1. We use spectral normalization Miyato et al. (2018) to satisfy the Lipschitz constraint: we replace the weight matrix $w$ of the convolution kernel by $w_{SN}(w) := c \times w/\sigma(w)$ where $\sigma(w)$ is the spectral norm of $w$ and $c \in (0, 1)$. This guarantees that $f_\psi$ is invertible – see Behrmann et al. (2018).

The encoder and decoder are 4-layer convolutional neural net with batch norm (Ioffe & Szegedy (2015)). Both encoder and decoder used fully convolutional architectures with 5x5 convolutional

filters and used vertical and horizontal strides 2 except the last deconvolution layer we used stride 1. The injective function $f$ is a 1 layer convolutional network with 3*3 kernl and stride 1. $\text{Conv}_k$ stands for a convolution with $k$ filters, $\text{DeConv}_k$ for a deconvolution with k filters, BN for the batch normalization Ioffe & Szegedy (2015), Relu for the rectified linear units, and $\text{FC}_k$ for the fully connected layer mapping to $R^k$.

$$x \in R^{64 \times 64 \times 3} \to \text{injective} f(\cdot) \in R^{64 \times 64 \times 3}$$
$$\to \text{Conv}_{128} \to \text{BN} \to \text{Relu}$$
$$\to \text{Conv}_{256} \to \text{BN} \to \text{Relu}$$
$$\to \text{Conv}_{512} \to \text{BN} \to \text{Relu}$$
$$\to \text{Conv}_{1024} \to \text{BN} \to \text{Relu} \to \text{FC}_{100}$$

$$z \in R^{100} \to \text{FC}_{10 \times 10 \times 1024}$$
$$\to \text{DeConv}_{512} \to \text{BN} \to \text{Relu}$$
$$\to \text{DeConv}_{256} \to \text{BN} \to \text{Relu}$$
$$\to \text{DeConv}_{128} \to \text{BN} \to \text{Relu} \to \text{DeConv}_3 \to sigmoid(\cdot)$$
$$\to \text{injective} f(\cdot) \in R^{64 \times 64 \times 3}$$

We use batch size 100 and latent dimension $z_{dim} = 100$ in all CelabA experiments. For the $\delta$-VAE with fixed spread, we use the fixed Gaussian noise with 0 mean and $(0.5)^2 I$ covariance. We train the model for 500 epochs using Adam optimizer with learning rate $1e^{-4}$. The learning rate decay with scaling factor 0.9 every 100000 iterations.

For the $\delta$-VAE with fixed spread We first train a $\delta$-VAE with fixed $f(x) = x$ and fixed Gaussian noise with 0 mean and $(0.5)^2 I$ diagonal covariance for 300 epochs, the learning rate decay with scaling factor 0.9 every 100000. Then we start iterative training by doing one step inner maximisation over the spread divergence's parameter $\psi$ using Adam optimizer with learning rate $1e^{-5}$ and one step minimization over the model parameter's $(\theta, \phi)$ using Adam optimizer for additional 200 epochs.

We can share the first 300 epochs between two models. When we sample form two models, we first sample from a 100 dimensional standard Gaussian distribution $z \sim N(0, I)$ and use the same latent code $z$ to get samples from both $\delta$-VAE with fixed and learned spread, so we can easily compare the sample quality between two models.

## J  WOODBERRY

When evaluating the log probability of this Gaussian, we use the Woodberry identity

$$\Sigma^{-1} = (I\sigma^2)^{-1} - (I\sigma^2)^{-1} L (I - L^T (I\sigma^2)^{-1} L)^{-1} L^T (I\sigma^2)^{-1}$$

so we only have to invert a $R \times R$ matrix. A similar trick is applied to calculate the log determinant:

$$\log \det(\Sigma) = \log \det(I + L^T (I\sigma^2)^{-1} L) + 2D \log \sigma.$$

The parameter $L$ is trained using reparameterization trick. When sampling $\xi \sim N(\mu, LL^\mathsf{T} + I\sigma^2)$, we first sample $z \in \mathbb{R}^R$ from $\mathcal{N}(z|0, I)$ and then sample noise $\epsilon \sim N(0, I\sigma^2)$ , thus a sample from $N(\mu, LL^\mathsf{T} + I\sigma^2)$ can be represented by $\xi = Lz + I\sigma\epsilon + \mu$.

## K  DETERMINISTIC LINEAR LATENT MODEL

Our aim here is to show how the classical deterministic PCA algorithm can be derived through a maximum-likelihood approach, rather than the classical non-probabilistic least-squares derivation. This is remarkable since the likelihood itself is not defined for this model.

For isotropic Gaussian observation noise with variance $\gamma^2$, the Probabilistic PCA model (Tipping & Bishop, 1999) for $X$-dimensional observations and $Z$-dimensional latent is

$$x = Fz + \gamma\epsilon, \quad z \sim N(0, I_Z), \quad \epsilon \sim N(0, I_X),$$
$$p_\theta(x) = \mathcal{N}\left(y|0, FF^\mathsf{T} + \gamma^2 I_X\right) \tag{59}$$

When $\gamma = 0$, the generative mapping from $z$ to $x$ is deterministic and the model $p_\theta(x)$ has support only on a subset of $\mathbb{R}^X$ and the data likelihood is in general not defined for $Z < X$.

In the following we consider general $\gamma$, later setting $\gamma$ to zero throughout the calculation. To fit the model to iid data $\{x_1, \ldots, x_N\}$ using maximum likelihood, the only information required from the dataset is the data covariance $\hat{\Sigma}$. For $\gamma > 0$, the maximum likelihood solution for PPCA is $F = U_Z \left(\Lambda_Z - \gamma^2 I_Z\right)^{\frac{1}{2}} R$, where $\Lambda_Z$, $U_Z$ are the $Z$ largest eigenvalues, eigenvectors of $\hat{\Sigma}$; $R$ is an arbitrary orthogonal matrix. Using spread noise $p(y|x) = \mathcal{N}\left(y|x, \sigma^2 I_X\right)$, the spreaded distribution is a Gaussian $\tilde{p}_\theta(y) = \mathcal{N}\left(y|0, FF^\mathsf{T} + (\gamma^2 + \sigma^2)I_X\right)$. Thus, $\tilde{p}_\theta(y)$ is of the same form as PPCA, albeit with an inflated covariance matrix. Adding Gaussian spread noise to the data also simply inflates the sample covariance to $\hat{\Sigma}' = \hat{\Sigma} + \sigma^2 I_X$.

Since the eigenvalues of $\hat{\Sigma}' \equiv \hat{\Sigma} + \sigma^2 I_X$ are simply $\Lambda' = \Lambda + \sigma^2 I_X$, with unchanged eigenvectors, the optimal deterministic ($\gamma = 0$) latent linear model has solution $F = U_Z \left(\Lambda'_Z - \sigma^2 I_Z\right)^{\frac{1}{2}} R = U_Z \Lambda_Z^{\frac{1}{2}} R$.

We have thus recovered the standard PCA solution; however, the derivation is non-standard since the likelihood of the deterministic latent linear model $\gamma = 0$ is not defined. Since classical deterministic PCA cannot normally be described in terms of a likelihood, the usual derivation of PCA is to define it as the optimal least squares reconstruction solution based on a linear projection to a lower-dimensional subspace, see for example Barber (2012). Nevertheless, using the spread divergence, we learn a sensible model and recover the true data generating process if the data were exactly generated according to the deterministic model.

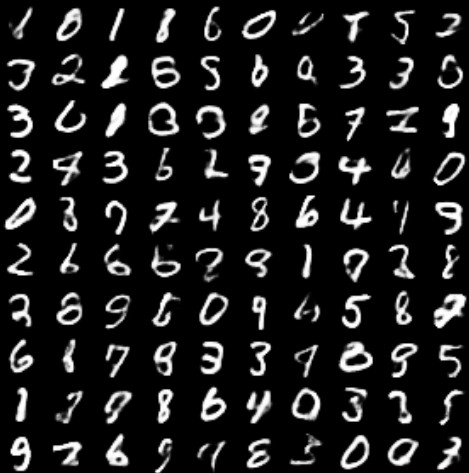

(a) Laplace with fixed covariance

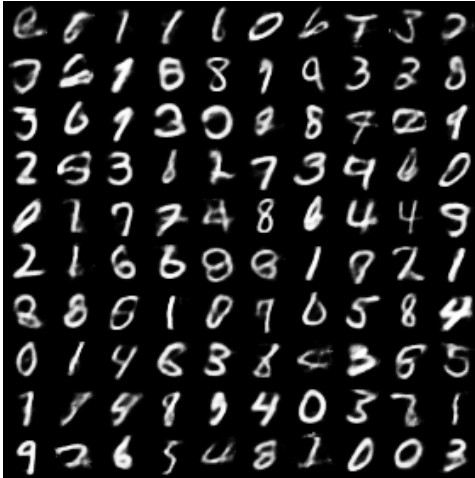

(b) Gaussian with fixed covariance

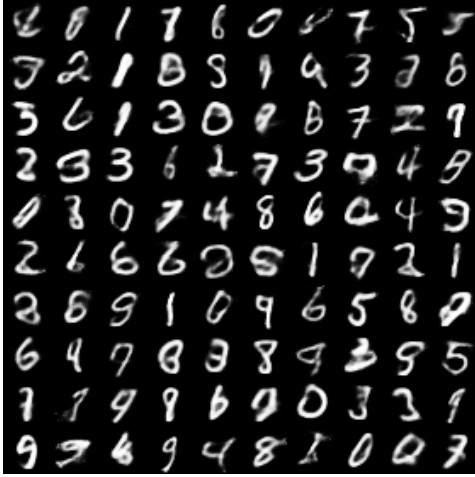

(c) Gaussian with learned covariance

Figure 4: Samples from a generative model (deterministic output) trained using $\delta$-VAE with (a) Laplace noise with fixed covariance, (a) Gaussian noise with fixed covariance and (c) Gaussian noise with learned covariance.

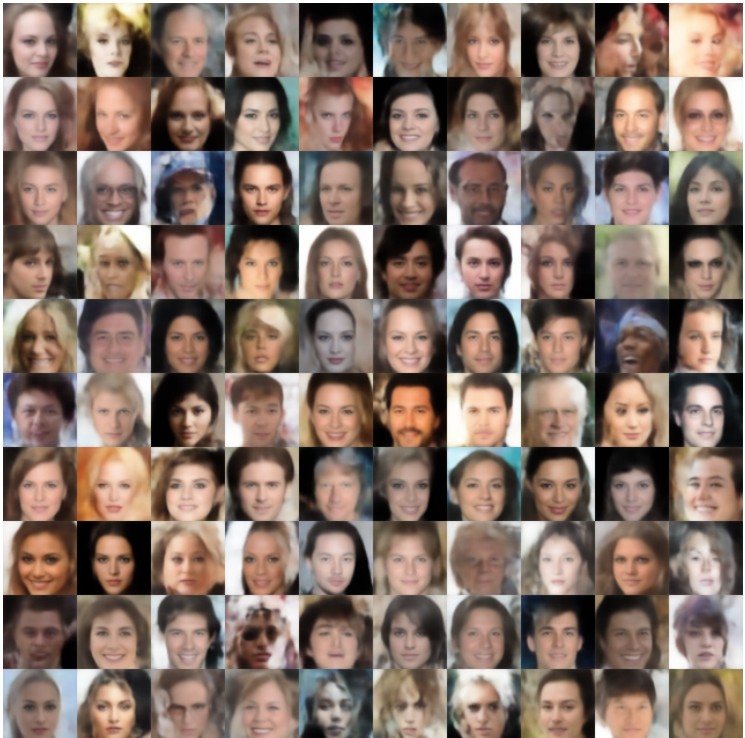

(a) Fixed spread noise

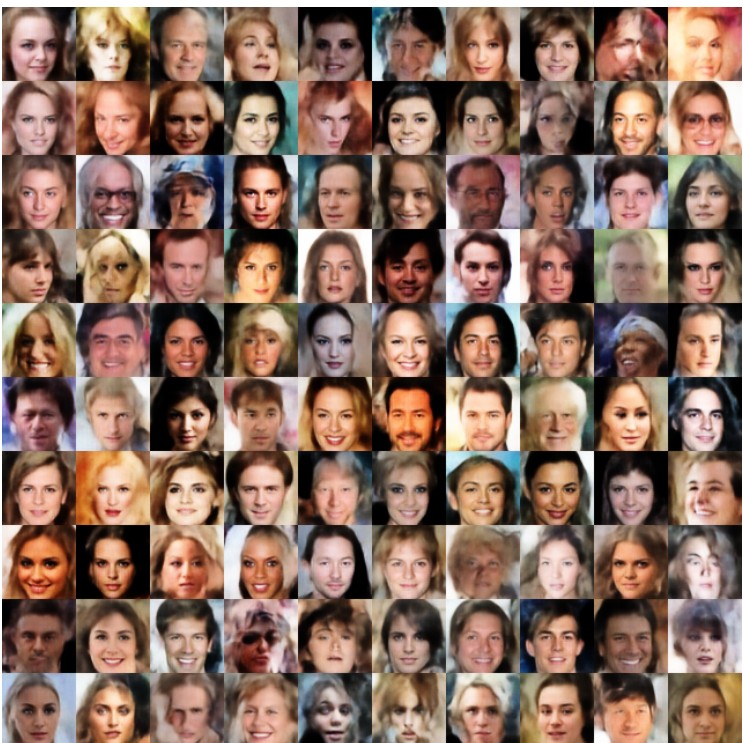

(b) Learned spread noise

Figure 5: Samples from a generative model with deterministic output trained using $\delta$-VAE with (a) fixed and (b) learned spread with injective mean transform.

