# OpenReview forum: "SPREAD  DIVERGENCE"
_ICLR.cc/2020/Conference — Reject_

### Official Review · AnonReviewer2 · 2019-10-21
**Official Blind Review #2**

**Rating:** 3

**Review:**

The paper introduced a way to modify densities such that their support agrees and that the Kullback-Leibler divergence can be computed without diverging. Proof of concept of using the spread KL divergence to ICA and Deep Generative Models ($\delta$-VAE) are reported based on the study of spread MLE.

Comments:

In Sec 1, mention that f should be strictly convex at 1. Also mention
Jensen-Shannon divergence, a KL symmetrization, which is always finite
and used in GAN analysis.

In Sec 2, you can also choose to dilute the densities with a mixture:
(1-\epsilon)p+\epsilon noise.
Explain why spread is better than that? Does spreading introduce
spurious modes?, does it change distribution sufficiency?
(Fisher-Neymann thm)

In Formula 4, there is an error: missing denominator of \sigma^2. See
Appendix D too.

In footnote 4, page 8, missing a 1/2 factor in from of TV (that is upper
bounded by 1 and not 2)

KL is relative entropy= cross-entropy minus entropy. What about spread KL?
In general, what statistical properties are kept by using the spread?
(or its convolution subcase?)

Is spreading a trick that introduces a hyperparameter that can then be
optimized for retaining discriminatory power, or is there
some deeper statistical theory to motivate it. I think spread MLE should
be further explored and detailed to other scenarii.

Spreading can be done with convolution and in general by Eq.3:

Then what is the theoretical interpretation of doing non-convolutional
spreading?


A drawback is that optimization on the spread noise hyperparameter is
necessary (Fig 3b is indeed much better than Fig 3a).
Is there any first principles that can guide this optimization rather
than black-box optimization?

Overall, it is a nice work but further statistical guiding principles
or/and new ML applications of spread divergences/MLE will strengthen the
work.
The connection, if any, with Jensen-Shannon divergence shall be stated
and explored.

Minor comments:

In the abstract, state KL divergence instead of divergence because
Jensen-Shannon divergence exists always.


Typos:
p. 6 boumd->bound
Bibliography : Cramir->Cramer, and various upper cases missing (eg.
wasserstein ->Wasserstein)


**Experience Assessment:**

I have published in this field for several years.

**Review Assessment: Checking Correctness Of Derivations And Theory:**

I assessed the sensibility of the derivations and theory.

**Review Assessment: Checking Correctness Of Experiments:**

I assessed the sensibility of the experiments.

**Review Assessment: Thoroughness In Paper Reading:**

I read the paper at least twice and used my best judgement in assessing the paper.

---

> ### Author Response · Authors · 2019-11-14
> **Response to Reviewer 2**
>
> We thank the reviewer for valuable reviews. We believe that we could fully address the concerns with the following arguments.
>
> 1. "The issues about JS:
> In the case of two distributions that have disjoint support, the JS divergence is always a finite constant. We think this is ill-defined since it is not a valid measure of divergence between two distributions and is not useful for statistical inference.  It is a common belief for vanilla GAN (with JS divergence)  that the source of instability during training is due to the disjoint support of the two distributions. Other distances are proposed to mitigate this effect [1].
>
> 2. "compare with a mixture of noise"
> In the discrete noise case, it is known as the anti-freeze method [2]. It is a special case of spread divergence. As noted in section 2.1; because the linear operator is equivalent to the convolution.
> However, in the continuous case mixture is different than spread; we can see this in the MLE setting, where if the data distribution is a delta function on the data points, the density of the resulting mixture at that point is still infinite (infinity times a constant is still infinity) so the MLE is still ill-defined, whereas the density of the spreaded distribution (with convolution noise) at that point is finite, (delta distribution will become a gaussian distribution). Therefore, we argue that the spread divergence is superior.
>
> 3. "Does spreading introduce spurious modes? "
> In the discrete case, spread noise will add density to the area that has 0 probability, so it will create local modes and it is necessary to define a valid spread divergence (but not spurious).  Therefore, below we answer to “does the spread noise introduce spurious local modes in the continuous case?”
>
> No. For stationary noise (proposed in our paper), it will never introduce additional modes.
> We assume a density function f  is differentiable in neighbourhood of the local mode. Since the mode is the local stationary point of the density function, $f’=0$ in the mode position. (This can be generalized to the case that  $f$ is locally continuous in the local mode, so there exits a sub derivative which equals to 0.)
> To define a spread divergence, we convolve f by a stationary noise $g$ ($g>0$ everywhere by the requirement of spread divergence).  The derivative of the convoluted distribution is given by $(g*f)’=g*f’$  (differentiation property of convolution, $*$ means convolution here).  Since g>0 everywhere, so $g*f’=0$ if and only if $f’=0$ , therefore spread noise will never introduce modes. However, other noise such as “mixture noise” $(1-\epsilon)p+\epsilon noise$ may potentially introduce additional modes.
>
> 4. "does it change distribution sufficiency?"
> No. The spread noise family introduced in the paper is a bijective operator (one to one mapping). Therefore, according to Fisher-Neymann theorem, it will not change the sufficiency of data statistics.
>
> 5. "statistical properties of using spread KL and potential applications"
> The statistical properties for inference for spread MLE are discussed in section 5.1 and appendix E.1. Comparing to KL, spread KL maintains the asymptotic efficiency and consistency properties, but only needs weaker conditions.
> For potential ML applications;  We have added a discussion in the end of section 5.2.1 of the revised paper.
>
> 6. "non-convolutional spreading"
> We agree that non-convolutional spreading noise is interesting.  However, this cannot be implemented easily for continuous systems in cases where we cannot evaluate explicitly the likelihood of the model. This means that one cannot directly use that method to train continuous implicit models using a modified EM approach. We will leave the analysis of non-convolutional spreading to be future work.
>
>
> 7. " Any principles that can guide this optimization rather than black-box optimization?"
> Optimization of the spread noise hyperparameters is not necessary for simple problems - see section 5.2.1, we achieve significant improvement using spread divergence using a fixed spread distribution.
> We agree in higher dimensional, more difficult problems learning the spread noise can improve performance significantly.
> We provide a principled method in section 4 to learn the spread distribution in an online fashion that maximises the discriminatory power, which we do not consider a black-box optimisation technique. Similar techniques are widely used in the kernel domain (MMD).
>
> 8. "missing denominator of $\sigma^2$"
> Thanks for pointing out the small error. We have added the assumption $\sigma^2=0.5$ within the revised paper.
>
> 9. "definition of TV distance"
> Our definition of TV was up to a constant; we have clarified within the revised paper.
>
> 10. We thank the reviewer for pointing out the typos, which we have fixed.
>
> [1] M. Arjovsky et al.  https://arxiv.org/abs/1701.07875
> [2]  T. Furmston, D. Barber, https://pdfs.semanticscholar.org/2ab5/475f67f5bdb6d4e411b8d7f3c56185b51847.pdf

---

### Official Review · AnonReviewer1 · 2019-11-04
**Official Blind Review #1**

**Rating:** 1

**Review:**

I think the paper must be Desk-rejected as the identity of the authors was revealed.
This thing aside, the paper is an interesting contribution. The concept of spread divergence can be valuable in many context. The presentation is thorough and the theoretical part is correct. On the other hand, the examples are quite diverse and include a standard model (ICA) as well as modern deep generative models. Thus, it represents a valuable contribution worth of publication, if we ignore the identity revelation aspect.

**Experience Assessment:**

I have published in this field for several years.

**Review Assessment: Checking Correctness Of Derivations And Theory:**

I carefully checked the derivations and theory.

**Review Assessment: Checking Correctness Of Experiments:**

I carefully checked the experiments.

**Review Assessment: Thoroughness In Paper Reading:**

I read the paper thoroughly.

---

> ### Author Response · Authors · 2019-11-14
> **Thank you for the positive review. The desk reject suggestion is unfortunate.**
>
> Thank you for the positive review. The desk reject suggestion is unfortunate - in attempting to release the code in a timely fashion to support our submission, the GitHub repository link https://github.com/zmtomorrow/spread_divergence_public used was associated to a personal Github account. We argue that the Github user is relatively inactive and utilises an anonymous alias, therefore, it would be difficult to identify the researcher behind the account. We claim it is as difficult to identify the researcher/their affiliations, as trying to find the paper in the public domain (given some other papers are openly endorsed on twitter and arxiv during the review process). Given this, we hope it does not exclude our submission and we have provided a new anonymous link.

---

### Official Review · AnonReviewer3 · 2019-11-04
**Official Blind Review #3**

**Rating:** 3

**Review:**

The paper proposes a new divergence, called spread divergence, to distinguish probability models. The approach is motivated from the concern that traditional divergence such as f-divergence or KL divergence may not always exist, in which the spread divergence may be a substitute. Some empirical supports are provided for the proposed method. Below I will summarize my concerns.

1. The spread divergence is proven no larger than the traditional divergence, so the paper claims this as an advantage of using spread divergence. My question is that if KL or f-divergence of two probability models is infinity, which means they distinguish the models very well, whether a new method is necessary (though it may provide a finite value).

2. As a new method, it would be useful to thoroughly compare it with the traditional ones. There is a lack theoretical comparison with the KL or f-divergence. Some numerical examples are provided but seem not enough. For instance, the applications focus on the situations where likelihood is not defined, i.e., data are deterministic w/o observation noise. It is interesting to see other examples where likelihood is defined and how traditional methods perform.

3. Kernel based spread divergence has been a major focus of this paper. It is interesting to see which kernel maximizes the spread divergence. Section 3.2 considers Gaussian kernel. Is this an optimal option?

4. Section 5 compares EM and spread EM based on one experiment and claims the latter has smaller error.   Does the same conclusion holds true in other examples?

I believe the motivation of this paper is interesting. This would be a stronger paper if more theoretical and empirical analysis can be added.

**Experience Assessment:**

I have published one or two papers in this area.

**Review Assessment: Checking Correctness Of Derivations And Theory:**

N/A

**Review Assessment: Checking Correctness Of Experiments:**

I carefully checked the experiments.

**Review Assessment: Thoroughness In Paper Reading:**

I read the paper at least twice and used my best judgement in assessing the paper.

---

> ### Author Response · Authors · 2019-11-14
> **Response to Reviewer 3**
>
> We thank the reviewer for valuable reviews. We believe that we could fully address the concerns with the following arguments.
>
> 1. In the case where the KL divergence is infinity between two probability models, it is not well defined in the sense it cannot be used for statistical inference. We illustrate this point in section 2 with the delta distribution example; the KL divergence will be infinity regardless of the location of the two distributions.
>
> 2.  Re the lack of theoretical comparison - the focus of the paper is on introducing a new divergence objective for applications of interest where distributions p and q have different support. From a theory perspective, we analyse how it relates to the widely used KL divergence as follows: In section 2, 3, we demonstrate how to augment the KL-divergence to produce the spread KL and under what sufficient conditions it is a valid divergence and will recover the true solution.
> In the MLE case, we compare and contrast KL and spread KL (see section 5.1 and appendix E). Our initial conclusion is that spread-KL preserves the favourable properties of MLE, under weaker conditions. Additional supporting theory can be future work and we believe we have provided a solid foundation, worthy of publication.
>
> Re the focus of the applications; situations where the likelihood is not defined, i.e., data are deterministic w/o observation noise - we argue that section 5.2.1, related to ICA, demonstrates exactly the comparison between spread divergence and the traditional MLE based EM algorithm, and how the performance is related to the amount of observation noise present. In figure 1. a, when observation noise is small, relative error explodes on the traditional methods (due to slow down/freezing), whereas our spread divergence method is unaffected.
>
> The set of problems of interest where the likelihood is not defined is large. In applications like deterministic ICA, maximum likelihood slow feature analysis, deterministic POMDP - EM cannot work effectively and heuristics exist with no guarantees. We believe the spread divergence provides a general framework which can be used to solve these problems in an elegant way.
>
> 3. The optimal choice of spread noise (kernel) will depend on the task. Hence, no, a Gaussian noise is not necessarily an optimal choice in general case.
>
> This choice is a hyper-parameter in our method - and setting parameters for a given choice of spread noise during training. We are restricted on which family of distributions we can use given our analysis in section 3 depends on the stationary characteristic for proving spread is a valid divergence. A more exhaustive ablation of noise choice is future work, for which we provide a solid foundation:
>
> We propose Gaussian and Laplace noise as convenient choices, which satisfy the stationary requirement and in practice were well behaved within our experiments. Empirically we compare the effectiveness of the Gaussian and Laplace choices on performance within section 5.2.2, where, qualitatively, we conclude that Laplace is a better choice (evidence to support that the Gaussian noise is a sub-optimal choice there).  See paper for intuition provided.
>
> For how to set the parameters for a given choice of spread noise, we provide a general strategy, which enables learning towards an optimal spread noise. The strategy is to maximize the power of a statistical test (or measure). For example, in the case of the delta-VAE in section 5.2.2, we maximise the discriminatory power of the spread divergence online wrt to the noise parameters, improving performance. Two complementary strategies are the Mean Transform and the Covariance Structure learning presented for the Gaussian case. These can be extended to other distributions, such as the Laplace, which have similar scale and location parameters.
>
> 4.  The same conclusion from the ICA experiment should hold for other problems where the slow down/freezing behaviour is present when an EM style likelihood approach is taken (even with the trick of adding small observation noise). This is caused when the posterior is not updated within the E-step. Other examples that we are aware of where this problematic phenomenon is present are policy learning in deterministic MDP/POMDP [2], maximum likelihood learning of SFA [slow feature analysis [3] (and other probabilistic matrix decomposition techniques). All of which are good candidates for a spread divergence application. Furthermore, the observation noise tricks of existing methods do not guarantee to recover the true data generating process, whereas our proposed spread divergence method can. We have added a point of clarification to section 5 of the paper.
>
> [1] M. Arjovsky et al., https://arxiv.org/abs/1701.07875
> [2] T. Furmston, D. Barber, https://pdfs.semanticscholar.org/2ab5/475f67f5bdb6d4e411b8d7f3c56185b51847.pdf
> [3] R. Turner et al.,http://www.gatsby.ucl.ac.uk/~turner/SFA/TSNCOMP2006v8.pdf

---

### Decision · Program_Chairs · 2019-12-19

**Decision:**

Reject

**Comment:**

This paper studies spread divergence between distributions, which may exist in settings where the divergence between said distributions does not. The reviewers feel this work does not have sufficient technical novelty to merit acceptance at this time.